# Corticosteroids May Have Negative Effects on the Management of Patients with Severe Fever with Thrombocytopenia Syndrome: A Case–Control Study

**DOI:** 10.3390/v13050785

**Published:** 2021-04-28

**Authors:** Takeshi Kawaguchi, Kunihiko Umekita, Atsushi Yamanaka, Seiichiro Hara, Tetsuro Yamaguchi, Eisuke Inoue, Akihiko Okayama

**Affiliations:** 1Department of Rheumatology, Infectious Diseases, and Laboratory Medicine, Faculty of Medicine, University of Miyazaki, Miyazaki 889-1692, Japan; takeshi_kawaguchi@med.miyazaki-u.ac.jp (T.K.); okayama@med.miyazaki-u.ac.jp (A.O.); 2Department of Internal Medicine, Miyazaki Prefectural Miyazaki Hospital, Miyazaki 880-8510, Japan; ayaman_555@yahoo.co.jp; 3Department of Internal Medicine, Miyazaki Prefectural Nichinan Hospital, Miyazaki 887-0013, Japan; shara@pref-hp.nichinan.miyazaki.jp; 4Department of Internal Medicine, Miyazaki Prefectural Nobeoka Hospital, Miyazaki 882-0835, Japan; yamaguchi@pref-hp.nobeoka.miyazaki.jp; 5Showa University Research Administration Center, Showa University, Tokyo 142-8555, Japan; eisuke.inoue@med.showa-u.ac.jp

**Keywords:** severe fever with thrombocytopenia syndrome, viral hemorrhagic fever, prognostic factor, corticosteroid

## Abstract

Severe fever with thrombocytopenia syndrome (SFTS) is an emerging viral hemorrhagic fever in China, Korea, and Japan. To date, no standardized treatment protocol for SFTS has been established. Corticosteroids (CS) may be administered to patients with SFTS and hemophagocytic syndrome, but its effectiveness and safety are still debatable. We conducted a retrospective case series review at four medical facilities in Miyazaki, Japan. Based on the medical records, clinical data, including the patients background, symptoms, physical findings, laboratory data at initial presentation, treatment, and outcome, were compared between the CS-treated and the non-CS-treated group. A total of 47 patients with confirmed SFTS in each hospital were enrolled in this study; there were 14 fatal cases and 33 nonfatal cases. The case fatality ratio was 29.8%. After adjusting patients’ background by propensity score matching, the case fatality ratio was higher (*p* = 0.04) and complications of secondary infections, including invasive pulmonary aspergillosis, tended to be more frequent (*p* = 0.07) in the CS-treated group than in the non-CS-treated group. These data suggested that administration of CS to patients with SFTS should be carefully considered.

## 1. Introduction

Severe fever with thrombocytopenia syndrome (SFTS) is a fatal hemorrhagic condition that was first reported in China in 2011 [1] and has been reported primarily in China, South Korea, and Western Japan. The virus causing SFTS belongs to the genus *Banyangvirus* of the family *Phenuiviridae*, *Huaiyangshan bangyangvirus* and is generally called the SFTS virus (SFTSV). SFTSV infection has been one of the most important public health issues in endemic areas because of its high case fatality ratio, which was reported to be 10–20% in China and South Korea [2,3] and 27–31% in Japan [4,5]. The poor prognostic backgrounds for SFTS include advanced age, male sex [6], and underlying diseases, such as malignancies, diabetes mellitus, chronic viral hepatitis, and chronic obstructive pulmonary disease [5,7]. In addition, patients with central nervous system manifestations [4,8], high levels of alanine aminotransferase (ALT), lactate dehydrogenase (LDH), and creatine kinase (CK) [8], renal dysfunction [9], coagulopathy, high level viremia, and hypercytokinemia were reported to show poor prognosis [10]. To date, no standardized treatment protocol for SFTS has been established. A cross-sectional study in China did not show the effectiveness of ribavirin [11]. In Japan, a clinical study on 23 patients with SFTS reported a beneficial effect of favipiravir, which reduced mortality (17.4%), compared with reported in previous studies [12]. Supportive therapies for patients with SFTS include blood transfusion, renal replacement therapy, plasma exchange, and antibiotics. Several case reports have shown that corticosteroid (CS) administration was effective for patients with SFTS and hemophagocytic syndrome (HPS) [13]. However, a retrospective study conducted in South Korea found that the CS-treated group had a higher mortality rate than the non-CS-treated group [14]. Owing to some reports on secondary fungal infections after CS administration for several SFTS cases [15], the safety of CS in SFTS needs to be evaluated. In this study, we aimed to identify the prognostic factors of SFTS and to determine whether CS is beneficial or not in patients with SFTS.

## 2. Materials and Methods

### 2.1. Patients

This study used the data of the registry study on tick-borne infectious diseases in Miyazaki, Japan [16]. We conducted a retrospective case series review at four medical facilities in Miyazaki, Japan. Adult patients aged ≥20 years and who were diagnosed as SFTS at the University of Miyazaki Hospital, Miyazaki Prefectural Miyazaki Hospital, Miyazaki Prefectural Nichinan Hospital, and Miyazaki Prefectural Nobeoka Hospital from January 2008 to September 2020 were registered in this study. Reverse transcriptase–polymerase chain reaction was used to detect the presence of the SFTSV gene in blood samples [17]; this test was performed at the Miyazaki Prefectural Institute for Public Health and Environment (Miyazaki, Japan), which is the laboratory authorized by the local government to test for these infectious agents. Based on the medical records, we collected clinical data, including patient’s background, symptoms, physical findings, laboratory data at initial presentation, treatment, and outcomes. Patient background included the season of infection, age, sex, underlying disease, independence in activities of daily living, and duration from the onset of illness to the first hospital visit. Altered mental status was defined as a Glasgow coma scale score <15 or Japan coma scale >0. The laboratory parameters comprised complete blood count, chemistry, and coagulation system tests. With regards to quality control management, the clinical laboratory divisions of these institutions have periodically performed external quality control checks of one other. Therefore, the quality control of laboratory data has been maintained among these institutions. Outcomes included in-hospital death and complications of secondary infection.

### 2.2. Study Protocol

The observation period was the duration of hospitalization or outpatient visit. To identify poor prognostic factors, the baseline clinical and laboratory parameters, treatment, and outcomes, including complications of secondary infections, were compared between the fatal and nonfatal groups. The previously reported poor prognostic factors [4,5,6,8,9] were compared between the CS-treated and non-CS-treated groups. The propensity score (PS) was estimated to predict the probability of receiving CS treatment in each patient, using the following key variables: age, mental status changes, and levels of platelet, blood urea nitrogen (BUN), and prothrombin time-international normalized ratio (PT-INR). The key variables were selected from the comparative data of the clinical features between the CS-treated and non-CS-treated groups in this study and from the known poor prognosis factors. After PS matching adjustment of the differences in the clinical characteristics between the CS-treated and non-CS-treated groups, we compared the clinical characteristics between the CS-treated and non-CS-treated groups to evaluate the therapeutic effects, prognosis, and complications of CS therapy.

### 2.3. Statistical Analysis

We used Fisher’s exact test and the Mann–Whitney U test to compare the categorical and continuous variables, respectively, between the fatal and nonfatal groups and between the CS-treated and non-CS-treated groups. The PS was estimated using a multivariate logistic regression model with the key variables mentioned above. To compare the outcomes between the CS-treated and non-CS-treated groups, matching was performed with a 1:1 matching protocol and a caliper width equal to 1/4 of the standard deviation of the PS. The duration from SFTS onset to death was analyzed using Kaplan–Meier estimates and was compared between the groups. Cox proportional hazards model was used to investigate the hazard ratio of whether CS was a potential risk factor for fatal outcomes. All tests were two-tailed, and *p* < 0.05 was considered statistically significant. All statistical analyses were performed using EZR (Saitama Medical Center, Jichi Medical University, Saitama, Japan), which is a graphical user interface for R (version 4.0.4; The R Foundation for Statistical Computing, Vienna, Austria).

## 3. Results

### 3.1. Prognostic Factors for SFTS

A total of 47 patients (46 inpatients and 1 outpatient) with confirmed SFTS were enrolled in this study; there were 14 fatal cases (in-hospital death) and 33 nonfatal cases. The case fatality ratio was 29.8%. Mortality in the early phase within 14 days from onset was noted in 12 cases and was caused by shock, respiratory failure, disseminated intravascular coagulation, and multiorgan failure. On the other hand, two cases died in the late phase after 14 days from onset (on the 23rd and 39th days, respectively); both had invasive pulmonary aspergillosis (IPA).

The clinical characteristics of the patients are shown in Table 1. The median age [inter-quartile range] of the patients was 76 [67–80.5] years and tended to be higher in the fatal group than in the nonfatal group. The proportion of men among all patients was low at 42.9% but was higher in the fatal group at 71.4%, compared with that in the nonfatal group. The underlying diseases, duration before hospital visit, clinical symptoms, and physical findings were comparable between groups. Regarding laboratory data at initial presentation, the degree of leukopenia, thrombocytopenia, and levels of ALT, LDH, and CK were not different between the groups. On the other hand, the levels of hemoglobin, BUN, creatinine, C-reactive protein (CRP), PT-INR, and D-dimer were higher in the fatal group than in the nonfatal group.

Treatment and outcomes, including complications of secondary infections, were comparable between the fatal and nonfatal groups (Table 2). None of the cases received antiviral therapy, such as ribavirin and favipiravir, nor did they undergo plasma exchange. Compared with the nonfatal group, the fatal group was administered CS, received blood transfusions, and was treated with mechanical ventilation more frequently. Overall, the complication rate of secondary infections was as high as 29.8%. The nonfatal group included several severe cases. Two patients required mechanical ventilation and received CS therapy. One patient was admitted to the intensive care unit and required continuous renal replacement therapy and recovered without CS administration. Invasive pulmonary aspergillosis (IPA) developed in four cases (8.5%). Compared with the nonfatal group, the fatal group tended to have more frequent secondary infections, including bacterial pneumonia, IPA, and bacteremia (Table 2).

### 3.2. Impact of CS Therapy in Patients with SFTS

Table 3 shows the comparison of the prognostic factors of poor outcome of SFTS according to CS administration. Among 47 patients with SFTS, CS was administered during hospitalization in 25 patients and was not given in 22 patients. Most patients were administered methylprednisolone 0.5–1 g/day for 3 days or prednisolone 1 mg/kg/day for 3–5 days. Although no unified protocol for CS administration was established, CS therapy was initiated on the day of admission or within a few days of admission as treatment for impaired consciousness and cytopenia at any one of the institutions. Compared with the non-CS-treated group, the CS-treated group had more known poor prognostic factors (i.e., altered mental status, severe thrombocytopenia, or LDH elevation at initial presentation). The case fatality ratio was high, and complications of secondary infections were more frequent in the CS-treated group than in the non-CS-treated group. All cases with bacterial pneumonia and IPA were in the CS-treated group.

The results of the comparison of the clinical characteristics and outcomes after PS matching between the CS-treated and non-CS-treated groups are shown in Table 4. A total of 24 patients in 12 pairs were matched, and the characteristics of cases in the CS-treated and non-CS-treated groups were adjusted, but some factors were not well balanced. The case fatality ratio was higher, and complications of secondary infections tended to be more frequent in the CS-treated group than in the non-CS-treated group.

### 3.3. Survival Time of the CS-Treated and the Non-CS-Treated Group

In the analysis before PS match, survival was significantly shorter in the CS group than in the non-CS-treated group (*p* = 0.02) (Figure 1A). Hazard ratio (HR) was 4.86 (95% confidence interval (CI): 1.08–21.88, *p* = 0.04). After PS matching, outcomes tended to be worse in the CS group than in the non-CS-treated group, although the difference was not statistically significant (*p* = 0.10) (Figure 1B), and HR was 3.44 (CI: 0.71–16.63, *p* = 0.13).

## 4. Discussion

In this study, the case fatality rate of SFTS was as high as 29.8% and was comparable with that in previous reports in Japan [4,5]. Of the 14 patients in the fatal group, 12 (85.7%) died within 14 days from onset, and the causes of death in the acute phase were consistent with those previously reported. The pathophysiology leading to poor prognosis in patients with SFTS probably includes HPS-associated cytokine storm, hemorrhagic tendency caused by thrombocytopenia and disseminated intravascular coagulation, and multiorgan failure [18]. The remaining two cases of death in the late phase were caused by secondary respiratory infection. Several studies have reported poor prognostic factors, such as patient background (age, sex, and underlying diseases), clinical symptoms, physical findings, and laboratory abnormalities. Similar to previous reports, this present study showed higher proportion of men and more severe renal dysfunction and coagulation abnormalities in the fatal group than in the nonfatal group. In addition, the hemoglobin levels were higher in the fatal group. As patients with SFTS have gastrointestinal symptoms such as loss of appetite, diarrhea, and renal dysfunction, blood concentration due to dehydration was assumed as the cause of the elevated hemoglobin levels. CRP was slightly high in mortality cases. We have previously reported that normal CRP level (<1 mg/dL) was characteristic of patients with SFTS [16]. The relative increase in the CRP value in the fatal group may have reflected cytokine storm and the complications of secondary infection, but the exact mechanism remains unknown. CS was administered to 53.2% of all cases; this rate was even higher at 85.7% in the fatal group. Other than our registry study on tick-borne infectious diseases in Miyazaki prefecture [16], there had been no report from Japan about the treatment components for SFTS. In China, the reported medications were ribavirin in 54–92% and CS in 20–24% [6,19,20]. Moreover, several studies from China and South Korea reported on the association of SFTS with IPA, which had a high complication rate of 20–31.9% [20,21]. The complication rate of IPA was lower in this present study than in the previous report from China, but the fatal group in this study may have included cases in which IPA could not be diagnosed.

In this study, CS was administered more frequently in the fatal group than in the nonfatal group. Based on our data, there seemed to be a tendency to administer CS to severe cases that had poor prognostic factors or had more severe cytopenia that was suggestive of HPS. Disease severity rather than the use of CS may have predisposed patients with SFTS to a high risk of death. However, even after adjusting the patient backgrounds, the case fatality ratio was significantly higher with CS use than without CS use. The mechanisms of worsening outcome with CS administration may include the adverse effect on host immunity, such as prolonged SFTSV RNA shedding and suppression of the host immune response to SFTSV. CS-induced complications, such as hyperglycemia, hypertension, and heart failure, may have affected prognosis, but these factors could not be examined in this study. The duration of CS administration was 3–7 days in most cases. Whether CS therapy was associated with poor prognosis in patients who died within a few days of administration of CS was not determined. In severe SFTS cases, SFTS could have rapidly progressed despite CS therapy. To the best of our knowledge, no consensus has been achieved on the duration of CS therapy in SFTS. CS therapy was continued depending on disease severity in most cases. Furthermore, the association between the duration of CS therapy and poor prognosis could not be evaluated in this study. Therefore, in future the association between the duration of CS therapy and the outcome of SFTS should be investigated.

Some severe cases that required mechanical ventilation or intensive care unit admission recovered after CS administration. Therefore, CS therapy may be useful in these serious cases. In some viral infections, CS administration has been reported to improve prognosis. Dokuzoguz et al. reported that concomitant use of CS with ribavirin was beneficial for patients with severe Crimean–Congo hemorrhagic fever [22], which is similar to SFTS in terms of virus characteristics, disease manifestations, and pathophysiology [18]. In recent events, coronavirus disease 2019 (COVID-19) has been found to affect host immunity processes, such as innate immune hyperactivation and adaptive immune dysregulation, resulting in worsening disease severity [23]. Although the role of CS in severe acute respiratory syndrome coronavirus 2 infections remains controversial, a British clinical trial has reported the efficacy of CS for treating COVID-19 [24]. A study in South Korea found that the survival time in the CS-treated group was shorter than that in the non-CS-treated group in mild cases (Acute Physiology and Chronic Health Evaluation II score <14) [14]. There had been no report to verify the effectiveness of CS for SFTSV infection in combination with ribavirin or favipiravir. Administration of CS to patients with SFTS should be carefully considered.

The increase in the rate of complications of secondary infections, especially IPA, with CS therapy in patients with SFTS remains unclear. In this study, there was a tendency for relatively frequent secondary infections, including IPA, in the CS-treated group. CS administration may affect host immunodeficiency and increase secondary infections. SFTSV infection itself was speculated to induce immunosuppression and lead to secondary infection as a result of leukopenia, thrombocytopenia, and interleukin 10 overexpression [21]. Liu et al. reported that the reduction of CD3^+^ and CD4^+^ T lymphocytes in patients with SFTS led to the suppression of immune function, which can increase the risk for secondary infection [25]. In fungal infections, Speth et al. described platelet activation, which resulted in fungal growth inhibition [26], thereby, implying that thrombocytopenia may promote fungal infection. Cunha et al. reported that the overexpression of interleukin 10 hampered the production of proinflammatory cytokines and the control of pathogen growth, which predisposed to invasive aspergillosis [27].

Our study had several limitations. First, the number of cases was small because only those that were registered in Miyazaki prefecture were included. The efficacy of CS administration should have been investigated by stratifying the severity of more patients. Moreover, some variables, such as the degree of mental status alteration, may have not been included in the PS matching between groups. PS matching could not fully adjust the differences in clinical characteristics between the CS-treated and non-CS-treated groups. Second, this study was retrospective in design. Although SFTS cases are often accompanied with hemophagocytosis, bone marrow puncture or biopsy was performed only in some cases in this study. A prospective study would be required to verify the efficacy and safety of CS. Third, this study did not assess cytokine kinetics and viral load to evaluate the effects of CS on SFTSV infection and host immunity. Unfortunately, with some cases, evaluation was only possible with stored samples, and we could not investigate the association between CS therapy and viral load, viral genome sequence, antibody response, or cytokine levels. Finally, we did not investigate the effect of CS in combination with antiviral agents, such as ribavirin and favipiravir.

## 5. Conclusions

Approximately one-third of patients with SFTS died and developed secondary infections, including IPA, during their hospital course. After PS matching, the case fatality ratio was higher and the complications of secondary infections, such as IPA, tended to be more frequent with CS use than without CS use. Administration of CS to patients with SFTS should be carefully considered and take into account the balance between therapeutic efficacy and adverse effects, which may depend on the dose, duration, and combination with antiviral medication.

## Figures and Tables

**Figure 1 viruses-13-00785-f001:**
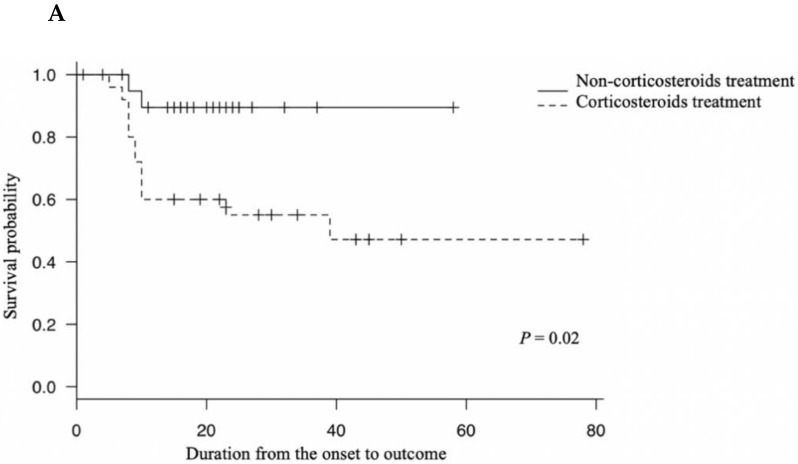
Kaplan–Meier curves for survival in the patients with severe fever with thrombocytopenia syndrome. (**A**) Survival curves in the corticosteroid-treated and the non-corticosteroid-treated groups (*p* = 0.02 by log-rank test). (**B**) Survival curves in the corticosteroid-treated and the non-corticosteroid-treated groups after propensity score matching (*p* = 0.10 by log-rank test).

**Table 1 viruses-13-00785-t001:** Clinical characteristics of the subjects.

	Total (*n* = 47)	Fatal (*n* = 14)	Nonfatal (*n* = 33)	*p* Value
Patients’ background				
Season				
Spring–Summer (March–August), *n* (%)	38 (80.9)	12 (85.7)	26 (78.8)	0.7
Autumn–Winter (September–February), *n* (%)	9 (19.1)	2 (14.3)	7 (21.2)	
Age, years	76 [67–80.5]	79.5 [76–82.9]	73.0 [67–78]	0.06
Male sex, *n* (%)	20 (42.6)	10 (71.4)	10 (30.3)	0.01
Underlying disease, *n* (%)	39 (83.0)	13 (92.9)	26 (78.8)	0.4
Hypertension, *n* (%)	21	7 (50.0)	14 (42.4)	0.89
Diabetes mellitus, (%)	5	2 (14.3)	3 (9.1)	0.84
Chronic viral hepatitis, *n* (%)	3	2 (14.3)	1 (3.0)	0.34
Chronic obstructive pulmonary disease, *n* (%)	2	1 (7.1)	1 (3.0)	0.51
Independence in activities of daily living, *n* (%)	46 (97.9)	14 (100)	32 (97.0)	1
Duration before hospital visit, days	4 [2–5]	4 [3–5]	4 [2–6]	0.56
Clinical symptoms and physical findings				
Fever, *n* (%)	45 (95.7)	14 (100)	31 (93.9)	1
Gastrointestinal symptoms, *n* (%)	42 (89.4)	13 (92.9)	29 (87.9)	1
Loss of appetite, *n* (%)	41 (87.2)	13 (92.9)	28 (84.8)	0.65
Diarrhea, *n* (%)	26 (55.5)	9 (64.3)	17 (51.5)	0.75
Neurologic symptoms, *n* (%)	27 (57.4)	9 (64.3)	18 (54.5)	0.75
Altered mental status, *n* (%)	24 (51.1)	8 (57.1)	16 (48.5)	0.75
Convulsion, *n* (%)	2 (4.3)	1 (7.1)	1 (3.0)	0.51
Hemorrhagic symptoms, *n* (%)	14 (29.8)	5 (35.7)	9 (27.3)	0.73
Laboratory data at initial presentation				
White blood cell count, /μL	1450 [1150–2000]	1310 [800–1560]	1600 [1220–2060]	0.06
Neutrophils, /μL	980 [530–1210]	580 [400–1140]	990 [620–1210]	0.27
Lymphocytes, /μL	440 [290–670]	320 [260–410]	540 [300–700]	0.08
Monocytes, /μL	90 [30–170]	40 [30–120]	90 [40–200]	0.1
Hemoglobin, g/dL	14.1 [13.2–15.1]	14.9 [14.1–15.4]	13.9 [12.1–14.8]	0.02
Platelets, ×103/μL	5.8 [4.5–7.6]	5.8 [4.3–8.0]	5.8 [4.6–7.4]	0.89
Alanine aminotransferase, IU/L	80 [45–144]	62 [45–151]	81 [44–136]	0.83
Lactate dehydrogenase, IU/L	567 [390–1016]	637 [389–930]	567 [407–1045]	0.97
Creatine kinase, IU/L	427 [188–832]	421 [223–1892]	427 [196–775]	0.59
Blood urea nitrogen, mg/dL	18.3 [15.0–26.5]	25.8 [17.2–32.6]	16.6 [13.4–21.1]	0.01
Creatinine, mg/dL	0.8 [0.7–1.1]	1.2 [0.8–1.4]	0.8 [0.7–1.0]	0.01
C-reactive protein, mg/dL	0.19 [0.08–0.56]	0.68 [0.21–0.85]	0.13 [0.06–0.37]	<0.01
Prothrombin time-INR	1.04 [0.94–1.08]	1.08 [1.04–1.14]	0.99 [0.91–1.06]	0.03
Activated partial thromboplastin time, seconds	46.5 [40.7–50.7]	47.5 [44.6–55.7]	46.4 [38.4–49.6]	0.14
D-dimer, ng/mL	6.9 [3.1–13.1]	11.0 [7.7–18.5]	5.1 [2.3–12.4]	0.04

All laboratory data were evaluated upon initial visit at our hospital. Data are presented as the median [inter-quartile range], unless otherwise specified. Patients with malignancies, end-stage renal disease, post-organ transplantation, or receiving immunosuppressive therapy were not included in both fatal and nonfatal groups. One patient in the fatal group and two in the nonfatal group had a history of malignancy. Hemorrhagic symptoms include petechiae, purpura, oral bleeding, and melena.

**Table 2 viruses-13-00785-t002:** Treatment and outcome including complication of secondary infections.

	Total (*n* = 47)	Fatal (*n* = 14)	Nonfatal (*n* = 33)	*p* Value
Treatment				
Antibiotics	32 (68.1)	12 (85.7)	20 (60.1)	0.17
Corticosteroids	25 (53.2)	12 (85.7)	13 (39.4)	<0.01
Transfusion	13 (27.7)	7 (50.0)	6 (18.2)	0.04
Intensive care unit admission	5 (10.6)	3 (21.4)	2 (6.1)	0.15
Mechanical ventilation	7 (14.9)	5 (35.7)	2 (6.1)	0.02
Continuous renal replacement therapy	3 (6.4)	2 (14.3)	1 (3.0)	0.2
Secondary infections *	14 (29.8)	6 (42.9)	8 (24.2)	0.3
Bacterial pneumonia	6 (12.8)	2 (14.3)	4 (12.1)	1
Pulmonary aspergillosis	4 (8.5)	3 (21.4)	1 (3.0)	0.07
Bacteremia	4 (8.5)	2 (14.3)	2 (6.1)	0.57

Data are presented as the number (%) of patients. * Secondary infections included cellulitis, which occurred in only one case in the nonfatal group. There was one case of overlapping pneumonia and bacteremia.

**Table 3 viruses-13-00785-t003:** Clinical characteristics of the subjects according to CS use.

Variables	Total (*n* = 47)	CS (*n* = 25)	No CS (*n* = 22)	*p* Value
Age, years	76 [67–80.5]	77 [69–82]	72 [66.3–79]	0.14
Male, *n* (%)	20 (42.6)	12 (48.0)	8 (36.4)	0.56
Altered mental status, *n* (%)	24 (51.1)	19 (76.0)	5 (22.7)	<0.001
White blood cell count, /μL	1450 [1150–2000]	1400 [990–1860]	1580 [1250–2000]	0.7
Hemoglobin, /μL	14.1 [13.2–15.1]	14.2 [13.2–15.3]	14.1 [13.0–14.8]	0.99
Platelets, ×10^3^/μL	5.8 [4.5–7.6]	5.0 [3.4–6.5]	6.7 [5.5–8.0]	0.05
Lactate dehydrogenase, IU/L	567 [390–1016]	808 [453–1079]	476 [335–606]	0.05
Blood urea nitrogen, mg/dL	18.3 [15.0–26.5]	19.5 [16.4–33.8]	15.5 [13.0–21.9]	0.17
Creatinine, mg/dL	0.8 [0.7–1.1]	0.86 [0.67–1.25]	0.8 [0.7–1.0]	0.49
C-reactive protein, mg/dL	0.19 [0.08–0.56]	0.19 [0.08–0.75]	0.20 [0.07–0.42]	0.53
Prothrombin time-INR	1.04 [0.94–1.08]	1.04 [0.91–1.08]	1.06 [0.98–1.09]	0.27
Outcome				
Fatal, *n* (%)	14 (29.8)	12 (48.0)	2 (9.1)	<0.01
Infection, *n* (%)	14 (29.8)	12 (48.0)	2 (9.1)	<0.01
Bacterial pneumonia, *n* (%)	6 (12.8)	6 (24.0)	0 (0.0)	0.02
Pulmonary aspergillosis, *n* (%)	4 (8.5)	4 (16.0)	0 (0.0)	0.11
Bacteremia, *n* (%)	4 (8.5)	3 (12.0)	1 (4.5)	0.61

All clinical characteristics, physical findings, and laboratory data were evaluated upon initial visit at our hospital. Data are presented as median [inter-quartile range], unless otherwise specified. CS, corticosteroid.

**Table 4 viruses-13-00785-t004:** Clinical characteristics of the subjects according to CS use after propensity score matching.

Variables	Total (*n* = 24)	CS (*n* = 12)	No CS (*n* = 12)	Std Diff
Age, years	78 [68.8–80.3]	76.5 [67.8–79.5]	78.5 [72.8–81]	0.35
Male, *n* (%)	13 (59.1)	7 (58.3)	6 (50.0)	0.17
Altered mental status, *n* (%)	11 (50.0)	5 (50.0)	5 (41.7)	0.17
White blood cell count, /μL	1500 [1150–1970]	1430 [930–1640]	1580 [1320–2020]	0.17
Hemoglobin, /μL	14.8 [14.0–15.4]	15.0 [14.1–15.3]	14.4 [13.7–15.7]	0.07
Platelets, ×103/μL	5.9 [5.2–7.1]	5.9 [4.9–7.0]	5.9 [5.4–7.1]	0.14
Lactate dehydrogenase, IU/L	567 [365–1080]	671 [358–1080]	558 [417–1078]	0.06
Blood urea nitrogen, mg/dL	19.3 [15.7–27.4]	19.8 [16.3–30.2]	17.4 [14.3–25.7]	0.11
Creatinine, mg/dL	0.89 [0.77–1.23]	0.84 [0.77–1.13]	0.98 [0.74–1.15]	0.09
C-reactive protein, mg/dL	0.35 [0.13–0.76]	0.39 [0.11–0.84]	0.33 [0.15–0.44]	0.53
Prothrombin time-INR	1.05 [0.96–1.08]	1.05 [0.97–1.15]	1.06 [0.96–1.06]	0.12
Outcome				*p* Value
Fatal, *n* (%)	10 (41.7)	8 (66.7)	2 (16.7)	0.04
Infection, *n* (%)	7 (29.2)	6 (50.0)	1 (8.3)	0.07
Bacterial pneumonia, *n* (%)	1 (4.2)	1 (8.3)	0 (0.0)	1
Pulmonary aspergillosis, *n* (%)	4 (16.7)	4 (33.3)	0 (0.0)	0.09
Bacteremia, *n* (%)	2 (8.3)	1 (8.3)	1 (8.3)	1

All clinical characteristics, physical findings, and laboratory data were evaluated upon initial visit at our hospital. Data are presented as median [inter-quartile range], unless otherwise specified. A standardized difference (Std diff) of <0.1 suggests adequate variable balance after propensity matching. CS, corticosteroid.

## Data Availability

The data presented in this study are available upon request from the corresponding author. The data are not publicly available because of ethical restrictions.

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
