# Peer review of "Corticosteroids May Have Negative Effects on the Management of Patients with Severe Fever with Thrombocytopenia Syndrome: A Case–Control Study"

_viruses, 2021, doi:10.3390/v13050785_

Round 1

Reviewer 1 Report

In this paper, the authors conducted a retrospective study of SFTS patients to determine that corticosteroid (CS) therapy is beneficial or not. Since standard treatments for SFTS have not been established, examining the usefulness of CS therapy provides clinicians with meaningful information. This paper is well-written; however, in order for the readers to assess correctly the usefulness and toxicity of the CS therapy, the author should add some information about the enrolled patients and detailed analyses of the data to the manuscript.

Major comments

  1. As the authors mention in Line 242-247, in order to examine the usefulness of CS therapy, patients should be prospectively studied for the use of CS by the severity of the disease. If such a prospective analysis is difficult to do, I believe that detailed information on each case becomes more important. In Table 2, I think that the two patients in the Non-fatal group who entered the ICU or required mechanical ventilation were very serious and severe cases. Did these patients receive or did not receive CS therapy? If CS was used and the patient survived, I consider that administration of CS might have been useful in such most severe cases. If there are such patients, the author should clarify them and comment on the positive aspects of CS therapy in the discussion part.

  1. Table 3. Did all 12 patients who died in the CS-treated group die after completing the administration of CS? From Figure 1(B), it can be observed that there are some death cases in the CS-treated group on the 5th to 6th day from onset. According to Table 1, deceased patients arrived at the hospital on average 4 days after onset, therefore, these patients may have been given CS for only 1-2 days. Since patients who died very early in the initiation of CS therapy died from SFTS itself, is it really correct to assume that they died due to the adverse effects of CS? If such patients are included in the study, the authors should add comments about this point to the discussion part.

  1. The authors do not state about the consent of the patients. Please clarify if your study has been approved by the Ethics Committees of every institutions. And, please describe how the consent of the patients participating in the study was obtained.

Minor comments

  1. As the authors state, administration of CS in patients with SFTS may be given to hemophagocytic syndrome (HS). Since the presence or absence of HS is considered to be information that the readers want to know, please add the presence or absence of HS (or hemophagocytosis in the bone marrow) to Table 3 or the text, if possible.

  1. Did any patients receive anti-viral therapy (ribavirin, favipiravir, and so on) as a treatment for SFTS? If there is such a patient, please add it to Table 2.

  1. Table 3 and Table 4. “Lactate dehydrpgenase” is incorrect. Please correct to “Lactate dehydrogenase”.

Author Response

To the editor and reviewers:

We have changed the title from “Corticosteroids may have negative effects in ....” to “Corticosteroids may have negative effects on ....” in the revised manuscript.

In our original manuscript, we stated that several studies from China reported on the association of SFTS with IPA. In the revised manuscript, we have changed “China” to “China and South Korea” (page 8, line 217).

Reviewer Comments:
Reviewer 1:

Major comments

  1. As the authors mention in Line 242-247, in order to examine the usefulness of CS therapy, patients should be prospectively studied for the use of CS by the severity of the disease. If such a prospective analysis is difficult to do, I believe that detailed information on each case becomes more important. In Table 2, I think that the two patients in the Non-fatal group who entered the ICU or required mechanical ventilation were very serious and severe cases. Did these patients receive or did not receive CS therapy? If CS was used and the patient survived, I consider that administration of CS might have been useful in such most severe cases. If there are such patients, the author should clarify them and comment on the positive aspects of CS therapy in the discussion part.

Response

We agree with your suggestion and have described the severe cases in the nonfatal group in the Results section. The nonfatal group included several severe cases. Two patients required mechanical ventilation and received CS therapy. One patient was admitted to the intensive care unit and required continuous renal replacement therapy and recovered without CS administration (page 4, lines 142–146). In addition, we have added the comment on the positive aspects of CS therapy in the Discussion section. Some severe cases that required mechanical ventilation or intensive care unit admission recovered after CS administration. Therefore, CS therapy may be useful in these serious cases (page 8, lines 241–243).

  1. Table 3. Did all 12 patients who died in the CS-treated group die after completing the administration of CS? From Figure 1(B), it can be observed that there are some death cases in the CS-treated group on the 5th to 6th day from onset. According to Table 1, deceased patients arrived at the hospital on average 4 days after onset, therefore, these patients may have been given CS for only 1-2 days. Since patients who died very early in the initiation of CS therapy died from SFTS itself, is it really correct to assume that they died due to the adverse effects of CS? If such patients are included in the study, the authors should add comments about this point to the discussion part.

Response

We appreciate your valuable comments. Two patients died on the fifth and seventh days after onset, respectively. They were hospitalized on day 1 and 4, respectively, and CS was administered for 3 days or more. We have added the comment on the duration of CS administration in the Discussion section (page 8, lines 232–235).

  1. The authors do not state about the consent of the patients. Please clarify if your study has been approved by the Ethics Committees of every institutions. And, please describe how the consent of the patients participating in the study was obtained.

Response

We have included a statement on the approval by the Ethics Committees and the consent of the patients participating in the “Institutional Review Board Statement” and “Informed Consent Statement” sections of the original manuscript. In the revised manuscript, they are listed on page 9, lines 229–304.

Minor comments

  1. As the authors state, administration of CS in patients with SFTS may be given to hemophagocytic syndrome (HS). Since the presence or absence of HS is considered to be information that the readers want to know, please add the presence or absence of HS (or hemophagocytosis in the bone marrow) to Table 3 or the text, if possible.

Response

As pointed out, considering whether the cases were complicated with HS (or hemophagocytosis in the bone marrow) or not is important. Unfortunately, in this study, we retrospectively collected clinical information. Bone marrow puncture or biopsy was performed only in some cases. Therefore, we could not mention the presence or absence of HS. We have added this point as a limitation of the study (page 9, lines 276–278).

  1. Did any patients receive anti-viral therapy (ribavirin, favipiravir, and so on) as a treatment for SFTS? If there is such a patient, please add it to Table 2.

Response

We agree with your suggestion and have added a description of the use of antiviral therapy to the revised manuscript (page 4, lines 138–140).

  1. Table 3 and Table 4. “Lactate dehydrpgenase” is incorrect. Please correct to “Lactate dehydrogenase”. 

Response

Thank you for pointing out this error. We have corrected the spelling in the revised manuscript (Tables 3 and Table 4).

Reviewer 2 Report

The manuscript described a retrospective review of the treatments and outcomes of 47 SFTS cases in 4 medical facilities in the prefecture of Miyazaki, Japan. Comparisons were made on the medical records, clinical data including the patients background, symptoms, physical findings, laboratory data at initial presentation, treatment and outcome; with special focus on the differences between the corticosteroids treated and non-treated patients. The results suggested that t the case fatality ratio was higher and complications of secondary infections more frequent in the corticosteroid-treated group, such that the administration of corticosteroid to patients with SFTS should be cautioned.

Major comments:

  1. The major limitation of the manuscript is well stated in the discussion (Lines 241-250), My main concern is on how well the propensity score matching can adjust the patients’ characteristics such that CS-treated and untreated groups can be reasonably un-biasedly compared. The use of CS maybe necessary for those patients (as did the blood transfusion), which may pre-dispose major differences (either disease type or status or severity) of the two comparing groups. In addition, the timing/purpose of CS use was not specified. How significant the dose and length of use of such CS have on the immunity (or other mechanisms) to result in the seen mortality rate difference was not clear and not discussed. It is very desirable that the viral load be determined when designing the experiment.
  2. The CS-associated hematological or biochemical (enzyme) alterations did not seem to be apparent. Are there any two or more cases with very similar backgrounds that differs only in CS treatment? What might be the fatality rate difference for patients having initial high and low Leukocytes? Or from clinical viewpoints, for patients having different disease status using combined parameters?  These are something interesting for comparison or discussion.
  3. The effect of combinational use with antiviral agents (such as ribavirin and favipiravir) were not but should be investigated, are these information available in this study? Are there other therapeutic agents concurrently used in the studied patients and what are possible effects those medications might have on the mortality rate. As stated in the discussion, the
  4. Do the 4 medical institutions use the same method/reagent and equipment for hematological and biochemical analysis? Different machines can very well generate data that have certain degrees of variation or differences that prevent them from direct comparisons.
  5. Is there any explanation why Hgb is higher in fatal group?

Minor comments:

  1. Abbreviations should be made when they first appear in the text.
  2. CBC was performed but only Hgb data was shown.

Author Response

Reviewer 2:

Major comments

  1. The major limitation of the manuscript is well stated in the discussion (Lines 241-250), My main concern is on how well the propensity score matching can adjust the patients’ characteristics such that CS-treated and untreated groups can be reasonably un-biasedly compared. The use of CS maybe necessary for those patients (as did the blood transfusion), which may pre-dispose major differences (either disease type or status or severity) of the two comparing groups. In addition, the timing/purpose of CS use was not specified. How significant the dose and length of use of such CS have on the immunity (or other mechanisms) to result in the seen mortality rate difference was not clear and not discussed. It is very desirable that the viral load be determined when designing the experiment.

Response

As pointed out, adjusting the clinical characteristics of the CS-treated and the non-CS-treated groups is important. Unfortunately, as mentioned in the Results section (page 6, lines 171–173) and in the limitations of the study, the number of cases in this study was limited, and some variables may not have been suitably adjusted. To the best of our knowledge, although no unified protocol for CS administration was established, CS therapy was initiated on the day of admission or within a few days of admission as treatment for impaired consciousness and cytopenia at any one of the institutions. We have described this point in the revised manuscript (page 5, lines 158–160). In addition, no consensus has been achieved on the duration of CS therapy. In clinical practice, both the duration and dosage of CS was considered by physicians depending on disease severity. We have described this point in the revised manuscript (page 8, lines 235–240). No blood samples were available for PCR and antibody response analysis. We wish to investigate this point in a future study. We have added a description about the viral load in the revised manuscript (page 9, lines 280–283).

  1. The CS-associated hematological or biochemical (enzyme) alterations did not seem to be apparent. Are there any two or more cases with very similar backgrounds that differs only in CS treatment? What might be the fatality rate difference for patients having initial high and low Leukocytes? Or from clinical viewpoints, for patients having different disease status using combined parameters?  These are something interesting for comparison or discussion.

Response

Thank you for these insightful questions. As shown in Table 3, compared with the non-CS-treated group, the CS-treated group had more known poor prognostic factors (e.g., altered mental status, severe thrombocytopenia, or LDH elevation at initial presentation). Therefore, after propensity-score matching adjustment of the differences in the clinical characteristics between the CS-treated and non-CS-treated groups, we compared the clinical characteristics between the CS-treated and non-CS-treated groups to evaluate the therapeutic effects, prognosis, and complications of CS therapy. This is as described in the original manuscript. Regarding white blood cell count, nearly all SFTS cases presented with leukopenia with only a slight difference between the fatal and nonfatal groups. We did not stratify or compare patients by white blood cell count.

  1. The effect of combinational use with antiviral agents (such as ribavirin and favipiravir) were not but should be investigated, are these information available in this study? Are there other therapeutic agents concurrently used in the studied patients and what are possible effects those medications might have on the mortality rate. As stated in the discussion, the

Response

We agree with your suggestion and have added a description of the use of antiviral therapy and other therapies in the revised manuscript (page 4, lines 138–140).

  1. Do the 4 medical institutions use the same method/reagent and equipment for hematological and biochemical analysis? Different machines can very well generate data that have certain degrees of variation or differences that prevent them from direct comparisons.

Response

As per your comment, the four medical institutions do not use the same method and equipment for hematological and biochemical analyses. With regards to quality control management, the clinical laboratory divisions of these institutions have periodically performed external quality control checks of one another. Therefore, the quality control of laboratory data has been maintained among these institutions. We have mentioned this in the Materials and Methods section (pages 2, lines 78–81).

  1. Is there any explanation why Hgb is higher in fatal group?

Response

We have considered your question and have added an explanation of elevated Hgb in the fatal group in the revised manuscript (page 7, lines 206–209).

Minor comments

  1. Abbreviations should be made when they first appear in the text.

Response

Thank you for pointing this out. We have reviewed and revised abbreviation use accordingly in the revised manuscript (page 2, lines 46–47; page 4, lines 145–146).

  1. CBC was performed but only Hgb data was shown.

Response

We understand your concern. Unfortunately, we did not extract data on RBC counts and hematocrit levels during the study.

Reviewer 3 Report

Authors describe clearly in their manuscript a cohort 47 individuals with severe fever with thrombocytopenia syndrome. I appreciate all the details provided in the manuscript. Authors state clearly three limitations of the study. Due to small cohort, it is difficult to draw statistically significant conclusion between CS and non-CS treated patients.  The trend is clearly evident but there are many confounding variables, which may also drive the difference. I would request two additional pieces of information for the patients. One, detail description of the underlying diseases in Table 1.  Two, can you provide any viral copy information and/or viral genome sequence information for the patients?

I am sure that the authors are aware of a similar study recently published by South Korean authors Jung et al. in PLoS Negl Trop Dis in February 2021.  They do address the same question and come to similar conclusion with the authors.  This work should be referenced in the beginning of the manuscript and the statement “but no study has confirmed its effectiveness and safety” should be revised.

I recommend the manuscript for publication after revision have been made by the authors.

Author Response

Reviewer 3:

Comments

I would request two additional pieces of information for the patients. One, detail description of the underlying diseases in Table 1.  Two, can you provide any viral copy information and/or viral genome sequence information for the patients?

Response

We agree with this suggestion and have added detailed descriptions of the underlying diseases in Table 1 and an accompanying footnote in the revised manuscript. We also agree with your suggestion on providing viral copy and viral genome sequence information. Unfortunately, only a small number of blood samples were available for analysis of viral copy or viral genome sequence. Therefore, we could not evaluate them in this study. We wish to investigate the association between the response to CS therapy and viral markers such as viral load and genome sequence in a future study. We have added a description about the viral load in the revised manuscript (page 9, lines 280–283).

I am sure that the authors are aware of a similar study recently published by South Korean authors Jung et al. in PLoS Negl Trop Dis in February 2021.  They do address the same question and come to similar conclusion with the authors.  This work should be referenced in the beginning of the manuscript and the statement “but no study has confirmed its effectiveness and safety” should be revised.

Response

Thank you for your comment. We have corrected the manuscript section that was pointed out and have cited the reference recommended by you (page 1, line 22; page 2, lines 54–57; page 8, lines 252–254) (new reference #14).

Round 2

Reviewer 1 Report

Comments

The revised manuscript is almost appropriately amended.    

Please check and revise the following three minor points to make this manuscript acceptable.

  • Line 235. Please change “To the our best of ...” to “To the best of...”.
  • Line 237. Please change “Turhtemore” to “Furthermore”.
  • Line 280. What is “wiht”? Please amend the sentence.

Author Response

Thank you for pointing out this error. We have corrected the spelling in the revised manuscript (Page 8, lines 235 and 237, and Page 9, line 280).

Reviewer 2 Report

My concerns have mostly been addressed, although some can not be satisfactorily answered due to experimental design, they are discussed in the updated manuscript.

Author Response

We appreciate your valuable comments. The points that could not be dealt with in this research will be considered as future issues.